# On-the-Job Learning with Bayesian Decision Theory

**Keenon Werling**
Department of Computer Science
Stanford University
keenon@cs.stanford.edu

**Arun Chaganty**
Department of Computer Science
Stanford University
chaganty@cs.stanford.edu

**Percy Liang**
Department of Computer Science
Stanford University
pliang@cs.stanford.edu

**Christopher D. Manning**
Department of Computer Science
Stanford University
manning@cs.stanford.edu

## Abstract

Our goal is to deploy a high-accuracy system starting with zero training examples. We consider an on-the-job setting, where as inputs arrive, we use real-time crowdsourcing to resolve uncertainty where needed and output our prediction when confident. As the model improves over time, the reliance on crowdsourcing queries decreases. We cast our setting as a stochastic game based on Bayesian decision theory, which allows us to balance latency, cost, and accuracy objectives in a principled way. Computing the optimal policy is intractable, so we develop an approximation based on Monte Carlo Tree Search. We tested our approach on three datasets—named-entity recognition, sentiment classification, and image classification. On the NER task we obtained more than an order of magnitude reduction in cost compared to full human annotation, while boosting performance relative to the expert provided labels. We also achieve a $8\%$ $F_1$ improvement over having a single human label the whole set, and a $28\%$ $F_1$ improvement over online learning.

> *"Poor is the pupil who does not surpass his master."*
> *– Leonardo da Vinci*

## 1 Introduction

There are two roads to an accurate AI system today: (i) gather a huge amount of labeled training data [1] and do supervised learning [2]; or (ii) use crowdsourcing to directly perform the task [3, 4]. However, both solutions require non-trivial amounts of time and money. In many situations, one wishes to build a new system — e.g., to do Twitter information extraction [5] to aid in disaster relief efforts or monitor public opinion — but one simply lacks the resources to follow either the pure ML or pure crowdsourcing road.

In this paper, we propose a framework called *on-the-job learning* (formalizing and extending ideas first implemented in [6]), in which we produce high quality results from the start without requiring a trained model. When a new input arrives, the system can choose to asynchronously query the crowd on *parts* of the input it is uncertain about (e.g. query about the label of a single token in a sentence). After collecting enough evidence the system makes a prediction. The goal is to maintain high accuracy by initially using the crowd as a crutch, but gradually becoming more self-sufficient as the model improves. Online learning [7] and online active learning [8, 9, 10] are different in that they do not actively seek new information *prior* to making a prediction, and cannot maintain high accuracy independent of the number of data instances seen so far. Active classification [11], like us,

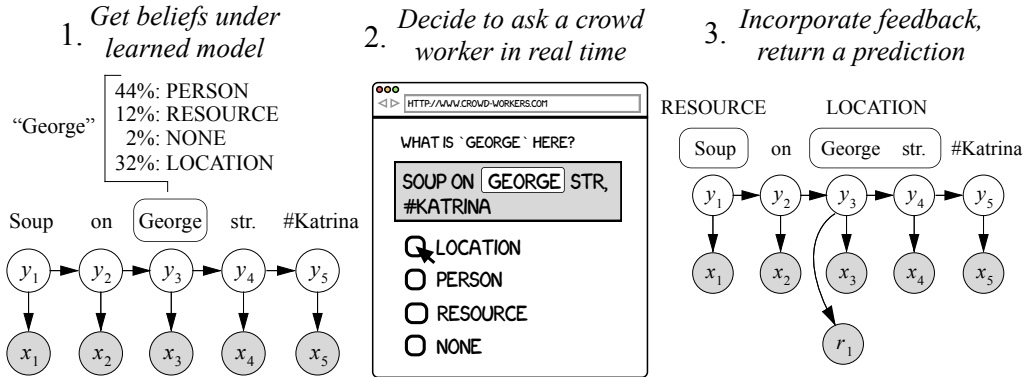

Figure 1: Named entity recognition on tweets in on-the-job learning.

strategically seeks information (by querying a subset of labels) prior to prediction, but it is based on a static policy, whereas we improve the model during test time based on observed data.

To determine which queries to make, we model on-the-job learning as a stochastic game based on a CRF prediction model. We use Bayesian decision theory to tradeoff latency, cost, and accuracy in a principled manner. Our framework naturally gives rise to intuitive strategies: To achieve high accuracy, we should ask for redundant labels to offset the noisy responses. To achieve low latency, we should issue queries in parallel, whereas if latency is unimportant, we should issue queries sequentially in order to be more adaptive. Computing the optimal policy is intractable, so we develop an approximation based on Monte Carlo tree search [12] and progressive widening to reason about continuous time [13].

We implemented and evaluated our system on three different tasks: named-entity recognition, sentiment classification, and image classification. On the NER task we obtained more than an order of magnitude reduction in cost compared to full human annotation, while boosting performance relative to the expert provided labels. We also achieve a 8% F1 improvement over having a single human label the whole set, and a 28% F1 improvement over online learning. An open-source implementation of our system, dubbed LENSE for "Learning from Expensive Noisy Slow Experts" is available at http://www.github.com/keenon/lense.

## 2 Problem formulation

Consider a structured prediction problem from input $\mathbf{x} = (x_1, \ldots, x_n)$ to output $\mathbf{y} = (y_1, \ldots, y_n)$. For example, for named-entity recognition (NER) on tweets, $\mathbf{x}$ is a sequence of words in the tweet (e.g., "*on George str.*") and $\mathbf{y}$ is the corresponding sequence of labels (e.g., NONE LOCATION LOCATION). The full set of labels of PERSON, LOCATION, RESOURCE, and NONE.

In the *on-the-job learning* setting, inputs arrive in a stream. On each input $\mathbf{x}$, we make zero or more queries $q_1, q_2, \ldots$ on the crowd to obtain labels (potentially more than once) for any positions in $\mathbf{x}$. The responses $r_1, r_2, \ldots$ come back asynchronously, which are incorporated into our current prediction model $p_\theta$. Figure 2 (left) shows one possible outcome: We query positions $q_1 = 2$ ("*George*") and $q_2 = 3$ ("*str.*"). The first query returns $r_1 = $ LOCATION, upon which we make another query on the the same position $q_3 = 3$ ("*George*"), and so on. When we have sufficient confidence about the entire output, we return the most likely prediction $\hat{\mathbf{y}}$ under the model. Each query $q_i$ is issued at time $s_i$ and the response comes back at time $t_i$. Assume that each query costs $m$ cents. Our goal is to choose queries to maximize accuracy, minimize latency and cost.

We make several remarks about this setting: First, we must make a prediction $\hat{\mathbf{y}}$ on each input $\mathbf{x}$ in the stream, unlike in active learning, where we are only interested in the pool or stream of examples for the purposes of building a good model. Second, the responses are used to update the prediction

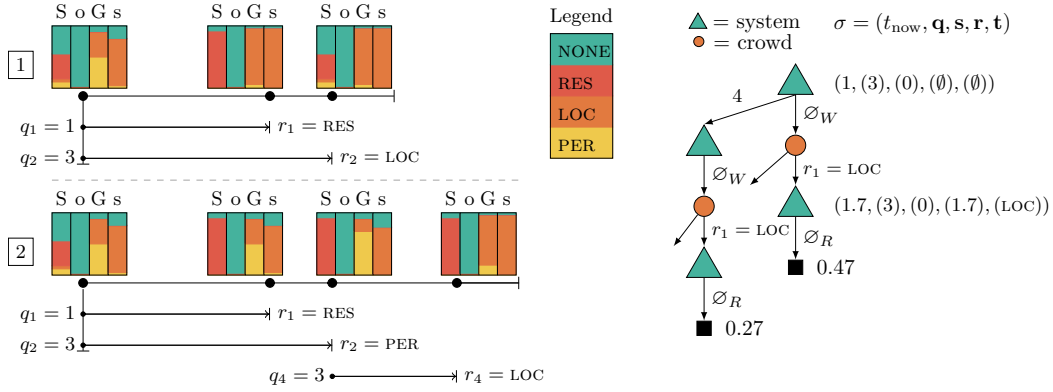

(a) **Incorporating information from responses.** The bar graphs represent the marginals over the labels for each token (indicated by the first character) at different points in time. The two time-lines show how the system updates its confidence over labels based on the crowd's responses. The system continues to issue queries until it has sufficient confidence on its labels. See the paragraph on behavior in Section 3 for more information.

(b) **Game tree.** An example of a partial game tree constructed by the system when deciding which action to take in the state $\sigma = (1, (3), (0), (\emptyset), (\emptyset))$, i.e. the query $q_1 = 3$ has already been issued and the system must decide whether to issue another query or wait for a response to $q_1$.

Figure 2: Example behavior while running structure prediction on the tweet "Soup on George str." We omit the RESOURCE from the game tree for visual clarity.

model, like in online learning. This allows the number of queries needed (and thus cost and latency) to decrease over time without compromising accuracy.

## 3 Model

We model on-the-job learning as a stochastic game with two players: the system and the crowd. The game starts with the system receiving input $\mathbf{x}$ and ends when the system turns in a set of labels $\mathbf{y} = (y_1, \ldots, y_n)$. During the system's turn, the system may choose a query action $q \in \{1, \ldots, n\}$ to ask the crowd to label $y_q$. The system may also choose the wait action ($q = \varnothing_W$) to wait for the crowd to respond to a pending query or the return action ($q = \varnothing_R$) to terminate the game and return its prediction given responses received thus far. The system can make as many queries in a row (i.e. simultaneously) as it wants, before deciding to wait or turn in.[1] When the wait action is chosen, the turn switches to the crowd, which provides a response $r$ to one pending query, and advances the game clock by the time taken for the crowd to respond. The turn then immediately reverts back to the system. When the game ends (the system chooses the return action), the system evaluates a utility that depends on the accuracy of its prediction, the number of queries issued and the total time taken. The system should choose query and wait actions to maximize the utility of the prediction eventually returned.

In the rest of this section, we describe the details of the game tree, our choice of utility and specify models for crowd responses, followed by a brief exploration of behavior admitted by our model.

**Game tree.** Let us now formalize the game tree in terms of its states, actions, transitions and rewards; see Figure 2b for an example. The *game state* $\sigma = (t_{\mathrm{now}}, \mathbf{q}, \mathbf{s}, \mathbf{r}, \mathbf{t})$ consists of the current time $t_{\mathrm{now}}$, the actions $\mathbf{q} = (q_1, \ldots, q_{k-1})$ that have been issued at times $\mathbf{s} = (s_1, \ldots, s_{k-1})$ and the responses $\mathbf{r} = (r_1, \ldots, r_{k-1})$ that have been received at times $\mathbf{t} = (t_1, \ldots, t_{k-1})$. Let $r_j = \emptyset$ and $t_j = \emptyset$ iff $q_j$ is not a query action or its responses have not been received by time $t_{\mathrm{now}}$.

During the system's turn, when the system chooses an action $q_k$, the state is updated to $\sigma' = (t_{\mathrm{now}}, \mathbf{q}', \mathbf{s}', \mathbf{r}', \mathbf{t}')$, where $\mathbf{q}' = (q_1, \ldots, q_k)$, $\mathbf{s}' = (s_1, \ldots, s_{k-1}, t_{\mathrm{now}})$, $\mathbf{r}' = (r_1, \ldots, r_{k-1}, \emptyset)$ and $\mathbf{t}' = (t_1, \ldots, t_{k-1}, \emptyset)$. If $q_k \in \{1, \ldots n\}$, then the system chooses another action from the new state $\sigma'$. If $q_k = \varnothing_W$, the crowd makes a stochastic move from $\sigma'$. Finally, if $q_k = \varnothing_R$, the game ends,

and the system returns its best estimate of the labels using the responses it has received and obtains a utility $U(\sigma)$ (defined later).

Let $F = \{1 \le j \le k - 1 \mid q_j \ne \varnothing_W \wedge r_j = \emptyset\}$ be the set of *in-flight* requests. During the crowd's turn (i.e. after the system chooses $\varnothing_W$), the next response from the crowd, $j^* \in F$, is chosen: $j^* = \arg\min_{j \in F} t'_j$ where $t'_j$ is sampled from the *response-time model*, $t'_j \sim p_{\mathrm{T}}(t'_j \mid s_j, t'_j > t_{\mathrm{now}})$, for each $j \in F$. Finally, a response is sampled using a response model, $r'_{j^*} \sim p(r'_{j^*} \mid \mathbf{x}, \mathbf{r})$, and the state is updated to $\sigma' = (t_{j^*}, \mathbf{q}, \mathbf{s}, \mathbf{r}', \mathbf{t}')$, where $\mathbf{r}' = (r_1, \ldots, r'_{j^*}, \ldots, r_k)$ and $\mathbf{t}' = (t_1, \ldots, t'_{j^*}, \ldots, t_k)$.

**Utility.** Under Bayesian decision theory, the *optimal choice* for an action in state $\sigma = (t_{\mathrm{now}}, \mathbf{q}, \mathbf{r}, \mathbf{s}, \mathbf{t})$ is the one that attains the maximum expected utility (i.e. value) for the game starting at $\sigma$. Recall that the system can return at any time, at which point it receives a utility that trades off two things: The first is the accuracy of the MAP estimate according to the model's best guess of $\mathbf{y}$ incorporating all responses received by time $\tau$. The second is the cost of making queries: a (monetary) cost $w_{\mathrm{M}}$ per query made and penalty of $w_{\mathrm{T}}$ per unit of time taken. Formally, we define the utility to be:

$$U(\sigma) \triangleq \mathrm{ExpAcc}(p(\mathbf{y} \mid \mathbf{x}, \mathbf{q}, \mathbf{s}, \mathbf{r}, \mathbf{t})) - (n_{\mathrm{Q}} w_{\mathrm{M}} + t_{\mathrm{now}} w_{\mathrm{T}}), \tag{1}$$

$$\mathrm{ExpAcc}(p) = \mathbb{E}_{p(\mathbf{y})}[\mathrm{Accuracy}(\arg\max_{\mathbf{y}'} p(\mathbf{y}'))], \tag{2}$$

where $n_{\mathrm{Q}} = |\{j \mid q_j \in \{1, \ldots, n\}\}|$ is the number of queries made, $p(\mathbf{y} \mid \mathbf{x}, \mathbf{q}, \mathbf{s}, \mathbf{r}, \mathbf{t})$ is a prediction model that incorporates the crowd's responses.

The utility of wait and return actions is computed by taking expectations over subsequent trajectories in the game tree. This is intractable to compute exactly, so we propose an approximate algorithm in Section 4.

**Environment model.** The final component is a model of the environment (crowd). Given input $\mathbf{x}$ and queries $\mathbf{q} = (q_1, \ldots, q_k)$ issued at times $\mathbf{s} = (s_1, \ldots, s_k)$, we define a distribution over the output $\mathbf{y}$, responses $\mathbf{r} = (r_1, \ldots, r_k)$ and response times $\mathbf{t} = (t_1, \ldots, t_k)$ as follows:

$$p(\mathbf{y}, \mathbf{r}, \mathbf{t} \mid \mathbf{x}, \mathbf{q}, \mathbf{s}) \triangleq p_\theta(\mathbf{y} \mid \mathbf{x}) \prod_{i=1}^{k} p_{\mathrm{R}}(r_i \mid y_{q_i}) p_{\mathrm{T}}(t_i \mid s_i). \tag{3}$$

The three components are as follows: $p_\theta(\mathbf{y} \mid \mathbf{x})$ is the *prediction model* (e.g. a standard linear-chain CRF); $p_{\mathrm{R}}(r \mid y_q)$ is the *response model* which describes the distribution of the crowd's response $r$ for a given a query $q$ when the true answer is $y_q$; and $p_{\mathrm{T}}(t_i \mid s_i)$ specifies the latency of query $q_i$. The CRF model $p_\theta(\mathbf{y} \mid \mathbf{x})$ is learned based on all actual responses (not simulated ones) using AdaGrad. To model annotation errors, we set $p_{\mathrm{R}}(r \mid y_q) = 0.7$ iff $r = y_q$,[2] and distribute the remaining probability for $r$ uniformly. Given this full model, we can compute $p(r' \mid \mathbf{x}, \mathbf{r}, q)$ simply by marginalizing out $\mathbf{y}$ and $\mathbf{t}$ from Equation 3. When conditioning on $\mathbf{r}$, we ignore responses that have not yet been received (i.e. when $r_j = \emptyset$ for some $j$).

**Behavior.** Let's look at typical behavior that we expect the model and utility to capture. Figure 2a shows how the marginals over the labels change as the crowd provides responses for our running example, i.e. named entity recognition for the sentence "Soup on George str.". In the both timelines, the system issues queries on "Soup" and "George" because it is not confident about its predictions for these tokens. In the first timeline, the crowd correctly responds that "Soup" is a resource and that "George" is a location. Integrating these responses, the system is also more confident about its prediction on "str.", and turns in the correct sequence of labels. In the second timeline, a crowd worker makes an error and labels "George" to be a person. The system still has uncertainty on "George" and issues an additional query which receives a correct response, following which the system turns in the correct sequence of labels. While the answer is still correct, the system could have taken less time to respond by making an additional query on "George" at the very beginning.

# 4 Game playing

In Section 3 we modeled on-the-job learning as a stochastic game played between the system and the crowd. We now turn to the problem of actually finding a policy that maximizes the expected utility, which is, of course, intractable because of the large state space.

Our algorithm (Algorithm 1) combines ideas from Monte Carlo tree search [12] to systematically explore the state space and progressive widening [13] to deal with the challenge of continuous variables (time). Some intuition about the algorithm is provided below. When simulating the system's turn, the next state (and hence action) is chosen using the upper confidence tree (UCT) decision rule that trades off maximizing the value of the next state (exploitation) with the number of visits (exploration). The crowd's turn is simulated based on transitions defined in Section 3. To handle the unbounded fanout during the crowd's turn, we use progressive widening that maintains a current set of "active" or "explored" states, which is gradually grown with time. Let $N(\sigma)$ be the number of times a state has been visited, and $C(\sigma)$ be all successor states that the algorithm has sampled.

---

**Algorithm 1** Approximating expected utility with MCTS and progressive widening

---

1: For all $\sigma$, $N(\sigma) \leftarrow 0$, $V(\sigma) \leftarrow 0$, $C(\sigma) \leftarrow [\,]$      ▷ Initialize visits, utility sum, and children
2: **function** MONTECARLOVALUE(state $\sigma$)
3:      increment $N(\sigma)$
4:      **if** system's turn **then**
5:          $\sigma' \leftarrow \arg\max_{\sigma'}\left\{\frac{V(\sigma')}{N(\sigma')} + c\sqrt{\frac{\log N(\sigma)}{N(\sigma')}}\right\}$      ▷ Choose next state $\sigma'$ using UCT
6:          $v \leftarrow$ MONTECARLOVALUE($\sigma'$)
7:          $V(\sigma) \leftarrow V(\sigma) + v$      ▷ Record observed utility
8:          **return** $v$
9:      **else if** crowd's turn **then**
10:          **if** $\max(1, \sqrt{N(\sigma)}) \leq |C(\sigma)|$ **then**      ▷ Restrict continuous samples using PW
11:              $\sigma'$ is sampled from set of already visited $C(\sigma)$ based on (3)
12:          **else**
13:              $\sigma'$ is drawn based on (3)
14:              $C(\sigma) \leftarrow C(\sigma) \cup \{[\sigma']\}$
15:          **end if**
16:          **return** MONTECARLOVALUE($\sigma'$)
17:      **else if** game terminated **then**
18:          **return** utility $U$ of $\sigma$ according to (1)
19:      **end if**
20: **end function**

---

# 5 Experiments

In this section, we empirically evaluate our approach on three tasks. While the on-the-job setting we propose is targeted at scenarios where there is no data to begin with, we use existing labeled datasets (Table 1) to have a gold standard.

**Baselines.** We evaluated the following four methods on each dataset:

1. **Human n-query**: The majority vote of $n$ human crowd workers was used as a prediction.

2. **Online learning**: Uses a classifier that trains on the gold output for all examples seen so far and then returns the MLE as a prediction. This is the best possible offline system: it sees perfect information about all the data seen so far, but can not query the crowd while making a prediction.

3. **Threshold baseline**: Uses the following heuristic: For each label, $y_i$, we ask for $m$ queries such that $(1 - p_\theta(y_i \mid \mathbf{x})) \times 0.3^m \geq 0.98$. Instead of computing the expected marginals over the responses to queries in flight, we simply count the in-flight requests for a given variable, and reduces the uncertainty on that variable by a factor of $0.3$. The system continues launching requests until the threshold (adjusted by number of queries in flight) is crossed.

| Dataset (Examples) | Task and notes | Features |
|---|---|---|
| **NER (657)** | We evaluate on the CoNLL-2003 NER task[3], a sequence labeling problem over English sentences. We only consider the four tags corresponding to persons, locations, organizations or none[4]. | We used standard features [14]: the current word, current lemma, previous and next lemmas, lemmas in a window of size three to the left and right, word shape and word prefix and suffixes, as well as word embeddings. |
| **Sentiment (1800)** | We evaluate on a subset of the IMDB sentiment dataset [15] that consists of 2000 polar movie reviews; the goal is binary classification of documents into classes POS and NEG. | We used two feature sets, the first (UNIGRAMS) containing only word unigrams, and the second (RNN) that also contains sentence vector embeddings from [16]. |
| **Face (1784)** | We evaluate on a celebrity face classification task [17]. Each image must be labeled as one of the following four choices: Andersen Cooper, Daniel Craig, Scarlet Johansson or Miley Cyrus. | We used the last layer of a 11-layer AlexNet [2] trained on ImageNet as input feature embeddings, though we leave back-propagating into the net to future work. |

Table 1: Datasets used in this paper and number of examples we evaluate on.

| System | Named Entity Recognition | | | | | Face Identification | | |
| | Delay/tok | Qs/tok | PER $F_1$ | LOC $F_1$ | ORG $F_1$ | $F_1$ | Latency | Qs/ex | Acc. |
|---|---|---|---|---|---|---|---|---|---|
| 1-vote | 467 ms | 1.0 | 90.2 | 78.8 | 71.5 | 80.2 | 1216 ms | 1.0 | 93.6 |
| 3-vote | 750 ms | 3.0 | 93.6 | 85.1 | 74.5 | 85.4 | 1782 ms | 3.0 | 99.1 |
| 5-vote | 1350 ms | 5.0 | **95.5** | 87.7 | 78.7 | 87.3 | 2103 ms | 5.0 | 99.8 |
| Online | n/a | n/a | 56.9 | 74.6 | 51.4 | 60.9 | n/a | n/a | 79.9 |
| Threshold | 414 ms | 0.61 | 95.2 | **89.8** | 79.8 | 88.3 | 1680 ms | 2.66 | 93.5 |
| **LENSE** | **267 ms** | **0.45** | 95.2 | 89.7 | **81.7** | **88.8** | 1590 ms | 2.37 | 99.2 |

Table 2: Results on NER and Face tasks comparing latencies, queries per token (Qs/tok) and performance metrics ($F_1$ for NER and accuracy for Face).

> Predictions are made using MLE on the model given responses. The baseline does not reason about time and makes all its queries at the very beginning.

4. **LENSE:** Our full system as described in Section 3.

**Implementation and crowdsourcing setup.** We implemented the retainer model of [18] on Amazon Mechanical Turk to create a "pool" of crowd workers that could respond to queries in real-time. The workers were given a short tutorial on each task before joining the pool to minimize systematic errors caused by misunderstanding the task. We paid workers $1.00 to join the retainer pool and an additional $0.01 per query (for NER, since response times were much faster, we paid $0.005 per query). Worker response times were generally in the range of 0.5–2 seconds for NER, 10–15 seconds for Sentiment, and 1–4 seconds for Faces.

When running experiments, we found that the results varied based on the current worker quality. To control for variance in worker quality across our evaluations of the different methods, we collected 5 worker responses and their delays on each label ahead of time[5]. During simulation we sample the worker responses and delays without replacement from this frozen pool of worker responses.

**Summary of results.** Table 2 and Table 3 summarize the performance of the methods on the three tasks. On all three datasets, we found that on-the-job learning outperforms machine and human-only

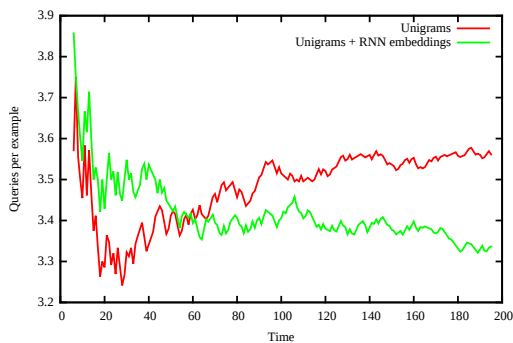

Figure 3: Queries per example for LENSE on Sentiment. With simple UNIGRAM features, the model quickly learns it does not have the capacity to answer confidently and must query the crowd. With more complex RNN features, the model learns to be more confident and queries the crowd less over time.

| System | Latency | Qs/ex | Acc. |
|---|---|---|---|
| 1-vote | 6.6 s | 1.00 | 89.2 |
| 3-vote | 10.9 s | 3.00 | 95.8 |
| 5-vote | 13.5 s | 5.00 | 98.7 |
| UNIGRAMS | | | |
| Online | n/a | n/a | 78.1 |
| Threshold | 10.9 s | 2.99 | 95.9 |
| **LENSE** | 11.7 s | 3.48 | 98.6 |
| RNN | | | |
| Online | n/a | n/a | 85.0 |
| Threshold | 11.0 s | 2.85 | 96.0 |
| **LENSE** | 11.0 s | 3.19 | 98.6 |

Table 3: Results on the Sentiment task comparing latency, queries per example and accuracy.

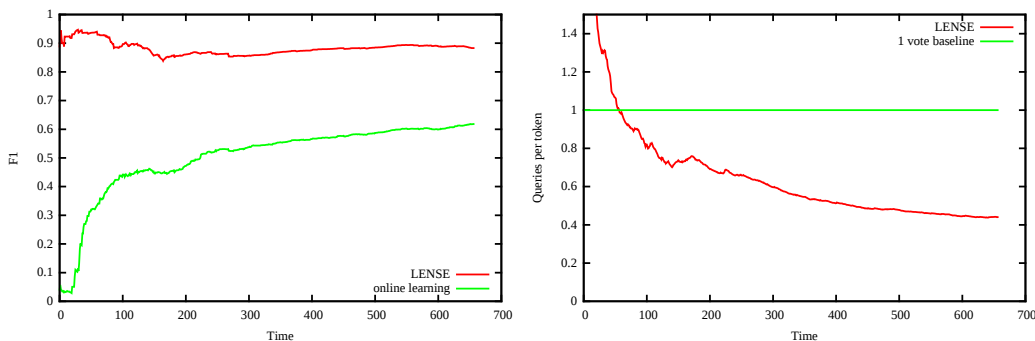

Figure 4: Comparing $F_1$ and queries per token on the NER task over time. The left graph compares LENSE to online learning (which cannot query humans at test time). This highlights that LENSE maintains high $F_1$ scores even with very small training set sizes, by falling back the crowd when it is unsure. The right graph compares query rate over time to 1-vote. This clearly shows that as the model learns, it needs to query the crowd less.

comparisons on both quality and cost. On NER, we achieve an $F_1$ of $88.4\%$ at more than an order of magnitude reduction on the cost of achieving comparable quality result using the 5-vote approach. On Sentiment and Faces, we reduce costs for a comparable accuracy by a factor of around 2. For the latter two tasks, both on-the-job learning methods perform less well than in NER. We suspect this is due to the presence of a dominant class ("none") in NER that the model can very quickly learn to expend almost no effort on. LENSE outperforms the threshold baseline, supporting the importance of Bayesian decision theory.

Figure 4 tracks the performance and cost of LENSE over time on the NER task. LENSE is not only able to consistently outperform other baselines, but the cost of the system steadily reduces over time. On the NER task, we find that LENSE is able to trade off time to produce more accurate results than the 1-vote baseline with fewer queries by waiting for responses before making another query.

While on-the-job learning allows us to deploy quickly and ensure good results, we would like to eventually operate without crowd supervision. Figure 3, we show the number of queries per example on Sentiment with two different features sets, UNIGRAMS and RNN (as described in Table 1). With simpler features (UNIGRAMS), the model saturates early and we will continue to need to query to the crowd to achieve our accuracy target (as specified by the loss function). On the other hand, using richer features (RNN) the model is able to learn from the crowd and the amount of supervision needed reduces over time. Note that even when the model capacity is limited, LENSE is able to guarantee a consistent, high level of performance.

**Reproducibility.** All code, data, and experiments for this paper are available on CodaLab at `https://www.codalab.org/worksheets/0x2ae89944846444539c2d08a0b7ff3f6f/`.

## 6 Related Work

On-the-job learning draws ideas from many areas: online learning, active learning, active classification, crowdsourcing, and structured prediction.

**Online learning.** The fundamental premise of online learning is that algorithms should improve with time, and there is a rich body of work in this area [7]. In our setting, algorithms not only improve over time, but maintain high accuracy from the beginning, whereas regret bounds only achieve this asymptotically.

**Active learning.** Active learning (see [19] for a survey) algorithms strategically select most informative examples to build a classifier. Online active learning [8, 9, 10] performs active learning in the online setting. Several authors have also considered using crowd workers as a noisy oracle e.g. [20, 21]. It differs from our setup in that it assumes that labels can only be observed *after* classification, which makes it nearly impossible to maintain high accuracy in the beginning.

**Active classification.** Active classification [22, 23, 24] asks what are the *most informative features* to measure at test time. Existing active classification algorithms rely on having a fully labeled dataset which is used to learn a static policy for when certain features should be queried, which does not change at test time. On-the-job learning differs from active classification in two respects: true labels are *never* observed, and our system improves itself at test time by learning a stronger model. A notable exception is Legion:AR [6], which like us operates in on-the-job learning setting to for real-time activity classification. However, they do not explore the machine learning foundations associated with operating in this setting, which is the aim of this paper.

**Crowdsourcing.** A burgenoning subset of the crowdsourcing community overlaps with machine learning. One example is *Flock* [25], which first crowdsources the identification of features for an image classification task, and then asks the crowd to annotate these features so it can learn a decision tree. In another line of work, *TurKontrol* [26] models individual crowd worker reliability to optimize the number of human votes needed to achieve confident consensus using a POMDP.

**Structured prediction.** An important aspect our prediction tasks is that the output is structured, which leads to a much richer setting for one-the-job learning. Since tags are correlated, the importance of a coherent framework for optimizing querying resources is increased. Making active partial observations on structures and has been explored in the measurements framework of [27] and in the distant supervision setting [28].

## 7 Conclusion

We have introduced a new framework that learns from (noisy) crowds *on-the-job* to maintain high accuracy, and reducing cost significantly over time. The technical core of our approach is modeling the on-the-job setting as a stochastic game and using ideas from game playing to approximate the optimal policy. We have built a system, LENSE, which obtains significant cost reductions over a pure crowd approach and significant accuracy improvements over a pure ML approach.

### Acknowledgments

We are grateful to Kelvin Guu and Volodymyr Kuleshov for useful feedback regarding the calibration of our models and Amy Bearman for providing the image embeddings for the face classification experiments. We would also like to thank our anonymous reviewers for their helpful feedback. Finally, our work was sponsored by a Sloan Fellowship to the third author.

## Footnotes

[1] This rules out the possibility of launching a query midway through waiting for the next response. However, we feel like this is a reasonable limitation that significantly simplifies the search space.

[2]We found the humans we hired were roughly 70% accurate in our experiments

[3]http://www.cnts.ua.ac.be/conll2003/ner/

[4] The original also includes a fifth tag for miscellaneous, however the definition for miscellaneos is complex, making it very difficult for non-expert crowd workers to provide accurate labels.

[5]These datasets are available in the code repository for this paper

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
