[Reviews · NeurIPS 2015]

Submitted by Assigned_Reviewer_1

The main contribution of this paper is to provide a more theoretically sound formulation of an on the job learner proposed in [22]. The learner starts by queuing the crowd to produce its output, but gradually starts making its own predictions and querying the crowd infrequently. The choice of whether to query the crowd for any given input is modeled as a decision problem solved via a Monte Carlo Tree Search.

The presentation is mostly good, although lacking in important details. The empirical results vis a vis a threshold-based heuristic are somewhat underwhelming. But it certainly seems like a reasonable approach and merits further investigation.

Please address/clarify the following:

1. Equation (2) isn't making much sense. The LHS should also be conditioned on s. And how does the y in p_{R}(r_i | y, q_i) vanish?

2. What is N(s) in Algorithm 1?

3. How did you get the 0.3 and 0.88 in the threshold baseline?

4. I am not familiar with the exact tasks used here, but are the "online" systems the best machine learned systems on this task? If not why not? If so, then any gain on top of it is purely as a result of human annotation. What are some such cases? Some error analysis would be handy.
Summary: This paper presents a theoretical formulation of an "on the job learner" that starts as a pure crowdsourcing system and gradually transitions to a learned model that only queries the crowd on low confidence inputs. The empirical results are somewhat of a mixed bag, but it will be good to get the discussion going along these lines.

Submitted by Assigned_Reviewer_2

I like that this paper compares classifier accuracy to human performance. It is interesting in Table 2 to see that the LENSE system outperforms the 3-vote baseline, but underperforms the 5-vote. This indicates a significant comparison about the relative efficiencies of de-noising crowd responses via additional consensus vs. the LENSE system approach. The example given in the paragraph starting at Line 154 is rather abstract and difficult to follow. For easy readability, it might be useful to use a specific query example with crowd responses and times similar to what they would be in a real case. In Tables 2 and 3, it is confusing that the threshold baseline is sometimes referred to as entropic and sometimes as threshold. I do not understand why the related work is in Section 5. Overall, a compelling problem but the explanations were somewhat opaque.

Line 086: Figure ??
Summary: This paper tackles the problem of creating high-accuracy classifiers beginning with zero labeled examples. A component problem in this paper is handling noisy crowd respondents, and while this topic has be considered from many angles in the literature, the timing and latency optimization factors of this paper are novel and interesting.

Submitted by Assigned_Reviewer_3

This paper presents a hybrid approach for using both

crowdsourced labels and an incrementally (online) trained model

to address prediction problems; the core idea is to lean heavily

on the crowd as the system is ramping up, learn from the labels

thus acquired, and then use the crowd less and less often as the

model becomes more confident. This is done via a sophisticated

framing of the problem as a stochastic game based on a CRF

prediction model in which the system and the crowd are both

players. The system can issue one or more queries q for tokens x

(with true label y) which elicit responses r, where there is a

utility U(q,r) for each outcome; the system thus attempts to

pick the actions that will maximize the expected

utility. Furthermore, the queries are not issued all at once,

but at times s (with response times t); utility is maximized

with respect to a t_deadline by which an answer needs to be

computed (this thus determines how many queries are sent out, at

what rate, etc.)

Computing this expected utility requires using

the simulation dynamics model P(y,r,t|x,q,s) in order to compute

the utilities as in (4). Given the utility values, the optimal

action could be chosen; however, the introduction of continuous

time makes this intractable to optimize and as such an

approximation is used based on Monte Carlo Tree Search and TD

learning (Algorithm 1). Experiments were conducted on a

simplified named entity recognition (NER) task, a sentiment

recognition task, and a face identification task, using four

methods: the majority vote of n human judges (1,3,5), online

learning (using the true labels), the "threshold" baseline (the

authors' model but without continuous time, in which m queries

are sent out at each step until the model prediction's

uncertainty is reduced below a threshold), and finally the

authors' full model (LENSE). In terms of precision/recall/F1,

the full model outperforms all but the 5-vote crowdsourced

version, though the "threshold" baseline does nearly as well

(and with lower latency for NER; Table 2). The authors also show

how the model requires fewer and fewer crowdsourced labels over

time given a sufficiently strong model to train (see Figure 3),

and that compared to online learning the accuracy is high from

the first example (Figure 4), since the system can leverage

crowd expertise more heavily while the model uncertainty is

still high.

This was a really interesting paper, and one that I expect could

generate both a lot of discussion and further work as well as

adoption in practice. There have been a variety of heuristic

approaches to learning from the crowd while training a system,

but this is the first complete and principled proposal I have

seen. The results against reasonable baselines are

impressive. As such, I feel the work is highly original and

significant, and in my mind deserves to be included in the

conference. That said, it does not fare nearly as well on the

clarity front, which unfortunately could detract from the

paper's potential impact. Many of these issues are correctible

(even for the camera-ready) if the authors are willing to put in

the time and effort and perhaps somewhat adjust the tone with

respect to the "threshold" baseline vs. the full continuous time

version.

The primary issue with clarity comes from the introduction of

continuous time, beginning partway through Section 3. The

motivation is reasonable, i.e., it might be necessary to operate

on a time (as well as a cost) budget for each classification,

but experimentally it seems to have very little benefit, while

the cost is that the formulation and discussion (as well as the

optimization procedure) become substantially more complex.

In

fact, in Table 2, it appears that the "threshold" baseline

achieves significantly lower latency on the NER task while still

getting roughly equivalent performance (marginally worse); on

the face ID task it actually performs better (though still

roughly equivalent). The authors argue (l. 314) that the

baseline does better on the second task because it's a single

label prediction task and doesn't have significant interaction

between labels - however, the NER task, which has such

interaction in spades, only seems to benefit marginally from

this.

For many practical tasks, the "threshold" baseline will

be good enough, and is already such a signficant gain (in terms

of being a principled hybrid learning system that outperforms

active and online learning).

The paper would likely have

greater impact if the authors made this clear in the

development, i.e., the core method could be developed fully

(without the introduction of time, and as such would be easier

to understand and implement), and the addition of continuous

time could be shown as an extension.

In the same vein, when

discussing the results, the authors could be more clear that the

performance of the "threshold" method is quite strong, even in

cases where there is a sequence dependence between tasks. This

is a suggestion and not strictly necessarily, as I feel the

paper is strong enough for inclusion as it is, but I do feel it

would improve the paper and the chances for the technique being

widely understood and adopted.

There are a number of smaller issues with respect to the results

and how they are reported. First, the introduction states that

the model "beats human performance" - but this is not true in

the 5-vote case for NER; strangely the 5-vote case is missing

for the Face-ID and Sentiment tasks in multiple tables and

figures (the right half of Tables 2, Table 3 and Figure 4) - it

really should be included. Likewise, in Table 2, the results for

the 5-vote case should be bolded as they represent the best

performance. The latency value for Threshold in NER should also

be bolded for the same reason (and the LENSE value for Face

ID). More importantly, the "Threshold" baseline is missing for

the sentiment experiment, and as such doesn't appear in Table 3

or Figure 4 - again, it should really be included as well.

A few minor comments with respect to the method description:

-While there is a discussion of t_deadline in l.183-187,

it is not clear what if any t_deadline was used in the

experiments, and what criteria was used to determine when the

answer was ready to report an answer - was it an entropic

threshold as with the baseline or was it reaching

some t_deadline?

-In equation (2), it seems the distribution needs to be given

the variable s as well (i.e., p(y,r,t|x,q,s).

-In equation 5, the use of q in F(q) is *very* confusing - I

assumed this referred to a query q, but in fact it just

represents a nominal distribution function - please choose some other

letter, as there are plenty to choose from.

-In the description of the Threshold baseline (l.246-252), I

assume the the expression should be (1 - p_theta(y_i|x))*0.3^m

<= (1-0.88) (note the RHS) as opposed to >= 0.88, as increasing

m will reduce uncertainty, not increase it.

Summary: This paper presents a hybrid approach for using both

crowdsourced labels and an incrementally (online) trained model

to address prediction problems, elegantly cast as a stochastic

game that models many aspects of the data, true labels, and

crowdworker responses, including the time at which queries are

sent out and answers received. Impressive results are shown with

respect to multiple human labelers, online learning with oracle

labels, and a "threshold" baseline using the authors' model but

removing the dependence on continuous time that performs on

par. The method is original, novel, and interesting, and as such

the paper should be accepted, but there are some issues with

clarity, particularly with respect to the introduction of

continuous time, which greatly complicates the discussion and

algorithmic mechanics yet seems to yield minimal benefits in

performance.

Submitted by Assigned_Reviewer_4

The authors describe a scenario called "on the job learning", where requests are made to annotators in between receiving an example and outputting a prediction.

Whilst it is clear how the algorithm works for a structured prediction task,

where elements of the sequence are queried, it's not clear at all how this was applied to the face recognition task - was it simply the instance being queried? In the non-structured setting the model wouldn't propagate information between adjacent positions (examples), so it seems like the utility function wouldn't make sense.

The threshold baseline has two parameters that are seemingly arbitrary. Where did these values come from and what is the sensitivity to these?

A further baseline of uncertainty sampling, akin to that used in active learning, would be interesting.

Finally, the authors are keen to make the distinction between this setting and active online learning, but active learning could be adapted quite simply to work in this domain, where the active set under consideration is now a sequence. For example, there's no reason why the model couldn't be updated in the same fashion on receiving responses, and then output a prediction at the end. Whilst this clearly wouldn't capture the temporal nature of the task, there is much theory to draw upon for choosing the query set.
Summary: A seemingly novel framework, although some details are not clear, and it seems to be fairly restricted in applicability.

Author Feedback
Author rebuttal: We would like to thank our reviewers for their thoughtful and thorough comments on our paper. In general, we are in complete agreement with the reviewers comments and will make all suggested corrections. We will also make an effort to revise to increase clarity throughout.

There were a few comments that we wanted to address further here:

Reviewer 7: "Active learning could be adapted quite simply to work in this domain, where the active set under consideration is now a sequence"
Reviewer 6: "comparisons to existing approaches lacking ... particularly with online AL algorithms."

The key challenge to applying AL here is that we are querying *in parallel*. While heuristics come to mind, adapting off-the-shelf AL to make principled decisions to query (or not) in the presence of uncertainty over the outcomes of several noisy in-flight queries on a variable is non-trivial. It's not as simple as providing an ordering over examples, since you need to take into account the queries already launched that have no response yet in providing that ordering. While we hope researchers apply the richness of AL literature to the on-the-job learning setting, it is a non-trivial adaptation, and out of scope for this paper.

Reviewer 5: Thank you for your very thoughtful comments. We'll address your points one-by-one. Everything we don't mention here (for space reasons) has still been heard, and will be addressed:
- "experimentally [Bayesian Decision theory] seems to have very little benefit [over the threshold baseline]"
The threshold baseline works well because it was carefully designed to approximate the Bayesian theory for simple problems at a much lower computational cost. In these experiments, the posterior after marginalization was relatively simple, so the approximation fits well. However, there are cases where the approximation breaks down. We will find space in the final version to walk through one.

- "For many practical tasks, the "threshold" baseline will be good enough"
With the caveat that "good enough" is still not optimal, we agree.

- "the NER task, which has such interaction in spades"
With the data sizes in our experiments (<1000 examples) the binary factors for an NER model are relatively weak. We suspect comparing LENSE and a threshold player on a model trained on the full CoNLL dataset would show a much more dramatic difference.

- "the "Threshold" baseline is missing for the sentiment experiment"
In an early draft we called the threshold baseline "Entropic," which we changed for reasons of clarity at the last minute, and forgot to relabel the sentiment table. This is a mistake and will be fixed.

- "strangely the 5-vote case is missing for the Face-ID and Sentiment tasks"
Time and budget reasons. We can have those experiments for the camera ready, if accepted.

- "it is not clear what if any t_deadline was used in the experiments, and what criteria was used to determine when the answer was ready to report an answer"
A timeout would confound comparisons with multi-vote systems that wait until for the last vote. Our game player decides when to turn-in (criteria for LENSE: the expected value of all queries is negative, and for Threshold: the confidence threshold is met for all variables). This was not sufficiently clear in the Model section, and will be revised.

Reviewer 2,7: "The threshold baseline has two parameters that are seemingly arbitrary. Where did these values come from and what is the sensitivity to these?"
These were chosen by hand on a dev set to provide the strongest possible baseline. The 0.3 is a rough approximation of the observed ~30% human error rate, and 0.88 best optimized the performance-cost tradeoff. The approx human error rate is insensitive, while minor changes to the threshold can cause dramatic jumps in the performance-cost tradeoff space. We will clarify this.

Reviewer 7: "it's not clear at all how this was applied to the face recognition task"
We treat this as a K-way classification task using the features obtained from the last layer of an 11-layer AlexNet, as we describe on lines 283-288. We will expand this explanation to avoid confusion.

Reviewer 2: "are the "online" systems the best machine learned systems on this task?"
The "online" system we compare against uses exactly the same model as LENSE, but trained with fully observed data instead of noisily and partially observed data that we get with human observations. It upper-bounds the achievable performance of LENSE without human help. Note any better model would also improve LENSE.

Reviewer 2: "What is N(s) in Algorithm 1?"
The number of times a state has been visited during sampling.

Reviewer 2: "Equation (2) isn't making much sense. The LHS should also be conditioned on s. And how does the y in p_{R}(r_i | y, q_i) vanish?"
You're right, the LHS of (2) is a typo. The y vanishes because it factorizes.